# Motivational Profile, Future Expectations, and Attitudes toward Study of Secondary School Students in Spain: Results of the PISA Report 2018

**DOI:** 10.3390/ijerph19073864

**Published:** 2022-03-24

**Authors:** Isabel Mercader Rubio, Nieves Fátima Oropesa Ruiz, Nieves Gutiérrez Ángel, Mª Mar Fernández Martínez

**Affiliations:** 1Department of Psychology, University of Almeria, 04120 Almería, Spain; foropesa@ual.es; 2Department of Sociology, Social Work and Public Health, University of Huelva, 21004 Huelva, Spain; mar.fernandez@dstso.uhu.es

**Keywords:** adolescence, self-determination, future expectations, motivation, PISA report

## Abstract

During the secondary school stage, students’ motivation to study may decrease and affect their future expectations, which are exclusively directed toward the search for employment, with the consequent abandonment of academic training. The main objective of the present paper was to examine the sources of motivation to study and the future expectations of secondary school students, as well as to develop a predictive model of their future expectations based on the variables studied. The sample consisted of a total of 35,943 students from different Spanish high schools, with an average age of 15.83 (SD = 0.28). The instrument used was the placement tests referring to the PISA 2018 report. On the one hand, the results showed that the main source of motivation for secondary school students to study responds to some kind of imposition either from the surrounding environment or internally, which appears to be represented by identified or controlled extrinsic motivation. In terms of future expectations, important factors included the fundamentally expression of their intention to continue studying rather than to stop studying, facts or economic support which they considered as influential to their capacity to study, and the opinions of others such as parents and friends. On the other hand, sex showed some significant differences in terms of future expectations but did not predict them. The regression model explained 20.9% of the variability of future expectations based on variables such as grade repetition, reasons that discourage studying (not being interested in the contents and never studying), and the influences on future expectations (school grades and subject mastery). Finally, the structural equation model revealed that grade repetition predicts the reasons that discourage studying and these in turn impact future expectations which are influenced by school grades, performing well in a specialty, and having talent. Likewise, there was a negative correlation between repeating a course and school grades, performing well in a specialty, and having talent. Based on these results, it would be advisable to improve the intrinsic motivation of secondary school students by means of educational actions that contribute to the adjustment of their future expectations and attend to the students’ own interests, desires, and competencies, all with the main purpose of contributing to meaningful learning and facilitating professional orientation, and above all, attending to diversity to reduce school failure.

## 1. Introduction

Quality in education and continuous learning are considered two fundamental ingredients in a society characterized by changes and where professional performance requires high and appropriate qualifications. During the secondary school stage, students experience multiple challenges: physical, psychological, social, and professional, and at the same time, this period is a window for learning. Paradoxically, the motivation to study in this period of development may decrease and the future expectations of young people may become exclusively oriented towards the search for employment, abandoning academic training. This is especially true for students with economic difficulties and a socio-family context with financial problems.

For these and other reasons, some studies have tried to develop educational strategies to improve learning motivation [1,2,3]. Within this framework, the self-determination theory (SDT) [4] corresponds to one of the psychological theories with the greatest globality and empirical support at present [5,6]. Within this theory, we find four sub-theories which attempt to provide an answer to the motives that lead a person to perform a certain activity: cognitive evaluation theory [7,8], organismic integration theory [4], causality orientation theory, and necessity theory [8]. In this paper, we focused on the causal orientations sub-theory. The main contribution of this theory corresponds to the definition of motivation as the reasons that direct our behavior, distinguishing between intrinsic and extrinsic motivation which are transformed as a function of self-determination, or in other words, the degree to which behavior is generated in the self [6].

In this sense, intrinsic motivation refers to innate motivation where we perform activities for the enjoyment it brings us [9], whereas extrinsic motivation refers to performing activities to obtain certain results, regardless of the task. We can distinguish three types of extrinsic motivation: integrated, identified, and controlled. Integrated motivation involves subjects combining different information which leads to greater cognitive involvement and learning. Identified motivation corresponds to the motivation that a subject feels with respect to an activity that is personally important to them. Controlled motivation is that in which the main reason is to obtain a reward, avoid punishment, or seek recognition from others [7]. 

Focused on the educational field, this contribution is linked to the reasons why students are or are not involved in certain school tasks. An example of intrinsic motivation is the need to achieve different academic achievements, while an example of extrinsic motivation is to obtain a certain grade in order to be able to access certain studies. Within this line of research, several studies have shown that it is most appropriate for students to develop a motivational process that starts with intrinsic motivation, passing through identified motivation and ending with controlled motivation [7,10,11]. With respect to the predominant type of motivation among secondary school students, several investigations with Spanish samples have shown that secondary school students are highly motivated [12,13].

Intrinsic motivation plays a relevant role in the personal and contextual adjustment of young people [14] and is associated with the development of emotional intelligence [15,16], improved attention to diversity [13], and higher academic performance [15,17]. Researchers have also been interested in exploring how the family context can contribute to school achievement. Thus, parental accompaniment, supervision, and support in the performance of school tasks have been correlated with higher levels of intrinsic motivation in secondary school students [18] while a lack of parental monitoring in this regard is considered a risk factor for positive academic performance in immigrant students, for example [19]. Other studies have focused on analyzing the role of certain sociodemographic variables such as sex and age in motivation. Regarding the relationship between sex and the motivation to study, in [20], significant sex differences were found with girls showing lower levels of extrinsic motivation than boys, although no significant differences were found in the intrinsic motivation of boys and girls. With regard to age, ref. [14] revealed that the level of motivation was higher in younger adolescents enrolled in the first years of compulsory secondary education.

We must not forget extrinsic motivation that focuses on the attribution of achievements to luck or chance and in which the driving force of the action depends on the reward to be achieved or the punishment to be avoided. However, this type of motivation also has a great educational value since it implies a preference for team tasks, peer tutoring, and in short, cooperative, collaborative, inclusive, and meaningful learning [20]. In addition, in adolescence, extrinsic motivation increases with respect to intrinsic motivation, or in other words, the interest of students is more focused on the achievement of specific goals and less on the aspects related to learning itself [21].

Since adolescence is a turning point in the construction of identity and personal projection where motivation plays a leading role, it seems essential to take into account the various motivational variables that influence academic performance in compulsory secondary education that are significant both for continuity and future choices of the student [22]. We studied these motivational factors and their relationship with repeating a year, similar to other studies with a similar purpose [23,24]. In the case of repeating students, there is a tendency to simplify the concept, uniting them within the same role or category despite a reality that is much more heterogeneous and complex. Studies such as the one elaborated by [25] established different types of repeaters, including within this classification the disciplined repeater, the rebellious repeater, the repeater with learning problems, etc. In this sense, it is essential to consider the results provided by the repeaters since they offer us a critical, non-judgmental, and evaluative vision of the teaching and learning processes.

What is certain is that repetition in compulsory secondary education is due to criteria related to school success or failure [26] highlighting that repetition is one of the more influential factors that negatively affects academic performance while repetition itself is enormously influenced by different social, school, or psychological factors (among which is motivation) [27,28,29,30]. Therefore, we cannot separate the family and social context from the school context, since it is in them that adolescents build their attitudes, personality, culture, and motivation [31]. Recent studies such as the one carried [32] have shown that such consequences are greater than the benefits and therefore, repetition has negative effects on students in both the short, medium, and long term [33] in terms of educational abandonment [34] or even negative repercussions at an emotional level [35]. However, other studies have shown that repetition can lead to an improvement in academic performance [32,36].

Another objective of this study was to analyze the relationship between the motivations that lead adolescents to study and their future expectations. However, studies that analyze this relationship with high school students are practically non-existent. In this sense, at a theoretical level, there are numerous contributions that have proven the influence of intrinsic motivation and academic performance, or in other words, there is currently clear evidence of the great influence that motivational variables have on academic performance [37,38,39,40]. However, we must not forget the influence that future expectations also have on academic performance: both academic and work-related [41], as well as the influence of parents [42] or the study habit itself [43].

Some other studies on future expectations in adolescence have focused mainly on the relationship between future expectations and life satisfaction, self-esteem, and self-efficacy [19] with a significant and positive relationship between these variables. On the other hand, less socially adjusted adolescents were found to have lower expectations [13], and the perception of lack of family support was also associated with lower future expectations [6] Regarding sex, some studies have found that boys showed lower future expectations than girls [13,44] and differences were also found in future expectations based on nationality and sex. Another factor was age [44], such that older adolescents showed lower academic and economic labor expectations than younger adolescents.

Based on the scientific literature, the objectives of this study were twofold: to study the participants’ sources of motivation for studying as well as their expectations for the future, and to identify possible differences that may exist in both variables according to sex, country of origin, languages spoken, and grade repetition. Likewise, the aim was to explore the relationship between attitudes toward studying and future expectations, and to develop a predictive and structural model for future expectations based on the variables studied, all regarding compulsory secondary education based on the data provided by the PISA 2018 report. Based on these objectives, the following hypotheses were formulated: (1) taking into account that most adolescents are guided by extrinsic motivation that varies across a continuum depending on the level of autonomy they have to act, we expect that the main source of motivation of the students is extrinsically internalized motivation and to a lesser extent intrinsic motivation; (2) there is a relationship between the type of motivation and the future expectations of high school students; (3) future expectations can be explained by variables such as sex, grade repetition, attitudes toward studying, and influences of other people or situations; and (4) it is possible to establish causal relationships between the above variables and draw a structural model for future expectations in adolescence.

## 2. Materials and Methods

On the one hand, contingency tables were used to measure the relationship between nominal variables, find the frequency distributions between two variables, and find statistical indices that measure the strength of the association between the variables (sex, grade repetition, attitudes toward studying, future expectations, and influences on expectations). On the other hand, a multinomial logistic regression analysis was used to explain the variability of a categorical variable with more than two levels such as future expectations based on qualitative independent variables (sex, grade repetition, attitudes toward studying, and influences on expectations). Finally, a structural equation model was used to estimate the causal relationships between exogenous variables (grade repetition and influences on future expectations) and endogenous variables (reasons for studying and future expectations).

### 2.1. Participants 

The sample was composed of a total of 35,943 students in compulsory secondary education from different Spanish high schools that participated in the 2018 PISA report. The mean age of the sample was 15.83 years with a standard deviation SD = 0.28. Regarding sex, 50% (*n* = 17,987) were boys and 50% (*n* = 17,956) were girls. As for repeaters, 82.9% (*n* = 29,129) had not repeated a grade and 11.8% (*n* = 6814) had repeated a grade. Regarding the country of origin, 88.8% (*n* = 31,901) were born in Spain while 9.1% (*n* = 3253) were born in a country other than Spain. In the case of the father’s country of birth, 81.7% (*n* = 29,381) were born in Spain while 14.9% (*n* = 5360) were born in a country other than Spain. In the case of the mother, 81.4% (*n* = 29,273) were born in Spain while 15.7% (*n* = 5653) were born in a country other than Spain. 

### 2.2. Instrument

The instrument used was the placement tests referring to the PISA 2018 report from which we chose only the items referring to questions of a sociodemographic nature (“Student Standardized Gender” and “Age”) and those related to the country of birth both of the participants and of their parents (e.g., “In what country were you and your parents born?”), whether they repeated a grade (“Grade Repetition”), the reasons for studying or not (“Why did you study before or after school?” with the options: I was interested in the content, we have a test coming up soon, my parents think studying is important, I had a homework assignment, all my classmates study before or after school, I always study, or other reason or “Why didn’t you study before or after school?” with the options: I was not interested in the content, there is no test coming up soon, I had no time to study, nobody told me I have to study, I had no homework assignment, none of my classmates study before or after school, I never study, or other reason), one’s own expectations for the future (“What do you think you will be doing 5 years from now?” with the options: I will be working because the occupation I want does not require a study degree, I will be working because I need to be financially independent, I will be studying because I do not know what I would like to do yet, I will be studying because the occupation I want requires a study degree, I will be studying or working for other reasons, or I will be doing something else), and influences on expectations (“Importance for decisions about future occupation” with the options: my parents’ or guardians’ expectations about my occupation, the plans my close friends have for their future, my school grades, the school subjects I am good at, my special talents, my hobbies, the social status of the occupation I want, financial support for education or training, education or training options for the occupation I want, employment opportunities for the occupation I want, or the expected salary of the occupation I want). Other studies that analyzed motivation in the educational field obtained acceptable reliability indices [45], same with the analysis of future expectations. The studies [19] showed adequate reliability indices.

### 2.3. Procedure

The results were collected and processed from the database corresponding to the answers of the PISA 2018 report level tests. The database is available on the website of the Ministry of Education and Vocational Training and the INEE (National Institute for Educational Evaluation). All materials and procedures were approved by the Bioethics Committee of the University of Almería.

### 2.4. Data Analysis 

Firstly, a descriptive statistical analysis was carried out for each dependent and relational variable using the Pearson chi-square test (χ^2^) (less than 20% of the cells had an expected frequency of less than 5). To verify the specific reason for the differences, if necessary, the corrected standardized residues were analyzed. In the cases in which the said value was greater than 1.96, it was where the differences occurred. To check the size of the effect, Cramer’s V was calculated. The coefficient obtained was interpreted as follows: <0.35 (small effect size), 0.35–0.65 (medium effect size), and >0.65 (large effect size) [46]. Secondly, to study the predictive capacity of different predictive variables (sex, grade or grade repetition, attitudes toward studying, and the influences on future expectations) on future expectations, a multinomial logistic regression analysis was carried out. Thirdly, to contrast the causal relationships between the variables, a structural equation modeling (SEM) was carried out. The estimation method used was maximun likelihood estimation (MLE). The quality of the estimated model was evaluated using different goodness-of-fit statistics that are generally used in the scientific literature. IFI (incremental fit index), TLI (Tucker Lewis index), and CFI (comparative fit index) were calculated with a parsimonious fit such as NFI (normed fit index), whose critical values are considered acceptable when they exceed 0.90 and optimal when they exceed 0.95 [47,48]. Other goodness-of-fit statistics such as RMSEA (room mean squared error of approximation) is considered acceptable if it is less than 0.8 and optimal if it is less 0.06 [47]. Statistical analysis was performed using SPSS version 27 and AMOS version 23.

## 3. Results

### 3.1. Motives That Do Not Encourage Study: Descriptive and Relational Analysis

Next, we analyzed the responses referring to the motives that lead students to study or not to study, taking into account differences in sex and whether or not they have been repeaters. Within this large section that we called motives are different variables related to the academic content itself such as having an exam coming up, having homework assigned, the parents’ idea of study, the fact that classmates also study, their own study habits, and lack of time.

Below are the most significant results of the chi-square test for the relationship between the reasons that do not encourage study and the degree of repetition for the variables that obtained a bilateral asymptotic significance value of less than 0.001 as well as the frequencies and their corrected residuals (Table 1).

The interpretation of the corrected residuals reflects that for repeating students, reasons that do not encourage them to study predominated, such as not being interested in the contents, that their classmates do not study, that nobody tells them that they have to study, and never studying.

### 3.2. Reasons for Study: Descriptive and Relational Analysis

In addition, the reasons that encouraged students to study in order of frequency were having an upcoming test with 87.1% (*n* = 23,550), having a homework assignment with 76.7% (*n* = 20,465), the fact that parents think that studying is important with 72% (*n* = 19,153), having the habit of always studying with 39.5% (*n* = 10,418), feeling interested in the contents with 37.7% (*n* = 9993), the fact that other classmates also study with 26.1% (*n* = 6848), and other reasons with 35.4% (*n* = 8583).

Table 2 specifies the most significant results of the chi-square test for the relationship between the reasons that encourage studying and the degree of repetition for the variables that obtained a bilateral asymptotic significance value of less than 0.001 as well as the frequencies and corrected residuals.

The interpretation of the corrected residuals reflects that for non-repeating students, reasons that encourage them to study predominated, such as being interested in the contents and their classmates studying, compared to repeating students who had a greater preoccupation in having a test soon (Table 2).

### 3.3. Short-Term Future Expectations: Descriptive and Relational Analysis

We also analyzed expectations regarding their own future in terms of the question “What do you think you will be doing in 5 years?” as well as the influence of agents such as parents, friends, academic performance, economic situation, social status, salary, or employment options on future expectations.

In order of frequency, 51.5% stated that they will be studying in five years because the position they intend to perform requires university credentials (*n* = 15,408), 13.6% stated the intention to continue studying because they do not yet know what they want to do (*n* = 4061), 11.8% intended to work because the job they want does not require university credentials (*n* = 3538), 10.6% intended to work because they want to be economically independent (*n* = 3174), 9.7% intended to study or work for other reasons (*n* = 2917), and 2.8% intended on doing something else (*n* = 835).

Pearson’s chi-square test showed statistically significant differences depending on grade repetition (χ^2^(5) = 3679.40; *p* = 0.000) with a medium effect size (V = 0.351). Looking at the contingency table, the group of repeaters had a greater presence in the following response options: working because the position they want to hold does not require university studies, working because they want to be financially independent, studying or working for other reasons, and doing something else; while non-repeating students had a greater presence in the option: studying because the position they intend to hold requires university studies, with their corrected residuals being higher than 1.96 in all cases (Table 3).

Table 4 shows the most significant results of the chi-square test for the relationship between future expectations and sex in the variables that obtained a bilateral asymptotic significance value of less than 0.001, as well as the corrected frequencies and residuals. (Table 4).

Pearson’s chi-square test showed statistically significant differences based on sex (χ^2^(5) = 720.62; *p* = 0.000) with a medium effect size (V = 0.155). Observing the contingency table, it can be seen that girls had a greater presence in future expectations related to continuing to study because the position they intend to hold requires university studies, while boys had a greater presence in the rest of the expectations: working because the position they want to hold does not require university studies, working because they want to be economically independent, studying because they still do not know what they want to do, studying or working for other reasons, or doing something else (Table 4).

### 3.4. Influence on Future Expectations: Descriptive and Relational Analysis

In addition, the sample of students indicated that the influence of other people or situations was significant for their future expectations. In order of frequency, 54.8% were influenced by the options of training or education (*n* = 14,005), 48.5% by being good at that specialty (*n* = 13,909), 47.3% by having talent for it, 46% by financial support (*n* = 12,979), 43% by grades (*n* = 12,380), 42.6% by social status (*n* = 1,212,124), 42.4% by their hobbies (*n* = 12,138), 41.7% by parents’ opinion (*n* = 12,083), and 25.6% by the plans their friends have (*n* = 7359).

Table 5 shows the most significant results of the chi-square test for the relationship between the influences on future expectations and the degree of repetition for those variables that obtained a bilateral asymptotic significance value of less than 0.001, as well as the corrected frequencies and residuals.

The contingency table shows that for non-repeaters, the influence of grades on expectations was very important, while for repeaters they ranged from having no importance, some importance, or importance. The same results were found for being good at a specialty. Regarding the influence of friends, the non-repeaters considered it to be important or very important to a greater extent, while the repeaters reported that it is somewhat important or not at all important. On the contrary, considering that having talent influences future expectations, repeating students considered this influence to be very important compared to non-repeating students who considered it either not important at all, somewhat important, or important.

### 3.5. Relationship between Attitudes toward Studying and Expectations for the Future

Regarding the relationship between the reasons that prompt students to not study and the future expectations of secondary school students, the Pearson chi-square test yielded statistically significant differences for the following reasons: not having time (χ^2^ = 11.20; *p* = 0.048), not being interested in the content (χ^2^ = 117.14; *p* = 0.000), no one telling them they have to study (χ^2^ = 119.36; *p* = 0.000), the fact that classmates do not study (χ^2^ = 103.72; *p* = 0.000), the fact that the student never studies (χ^2^ = 195.97; *p* = 0.000), and other reasons that do not incite them to study (χ^2^ = 45.93; *p* = 0.000). However, the test did not show statistically significant differences with regard to not having assigned homework and not having upcoming exams. 

Next, the most significant results of the chi-square test are presented for the relationship between the attitude towards the study and future expectations in those variables that obtained a bilateral asymptotic significance value of less than 0.001, as well as the frequencies and corresponding corrected residuals (Table 6).

Taking into account the contingency table, it can be seen that a greater proportion of students who were not interested in the contents reported future expectations related to working because the position they want to perform does not require university studies, working because they want to be economically independent, continuing studying since they still don’t know what they want to do, studying and working, and doing something else. The same was observed for the students who answered that no one tells them they have to study, as well as for those who studied because their classmates study (Table 6). A greater proportion of students who pointed out that they are discouraged to study because their classmates do not study reported future expectations related to working because the position they want to hold does not require university studies, working because they want to be economically independent, and studying and working. Similar results were obtained for students who never study, although in this case they were also doing something else.

### 3.6. Predictive Models for Future Expectations

With the aim of explaining future expectations, we selected variables that maintained a statistically significant relationship with this variable, taking into account the size of the effect as well as theoretical support for its exploration. For the multinomial regression analysis, we selected variables such as sex, grade repetition, reasons that do not encourage studying (not being interested in the contents and never studying), and influences on future expectations (school grades and subject mastery). The method chosen for the regression analysis was the main effects model.

The final model was significant χ^2^_(50)_ = 1042.79, *p* = 0.000. Pearson’s goodness-of-fit tests and deviance showed that the model fit well since the probability values were greater than 0.05, making it possible to reject the null hypothesis. The value of Nagelherke’s R2 was 209, which could be interpreted as meaning that 20.9% of the variability of future expectations could be explained through the variables introduced in the model. Table 7 shows the variables that have obtained a bilateral asymptotic significance value of less than 0.05 and that, therefore, explain the variability found in future expectations.

### 3.7. Structural Equation Model

The variables included in the model were the degree of repetition, reasons that do not encourage them to study (not being interested in the contents, that nobody tells them that they have to study, that their classmates do not study, and that they never study), future expectations and influences on future expectations (school grades, subject mastery, and being talented). Figure 1 presents the hypothesized predictive model as well as the causal relationships between the variables. The values of the goodness-of-fit indexes show an optimal fit: χ^2^ = 184.1, *p* = 0.000, IFI = 0.994, TLI = 0.985, CFI = 0.994, NFI = 0.994, and RMSEA = 0.017. The results of this model confirmed that grade repetition predicts the reasons that do not encourage studying and these in turn affect future expectations which are positively influenced by school grades, being good in a specialty, and having talent. Likewise, there was a negative correlation between grade repetition and the influences on future expectations (school grades, subject mastery, and having talent).

## 4. Discussion and Conclusions

The first objective of this work was to identify the sources of motivation that participants have towards studying. Therefore, this study shared the same purpose of studies examining the motivational profiles toward academic tasks. It is striking that most research on this subject focuses mainly on physical education [1,46,49], mathematics [3], or even the performing arts [2]. However, we have not found studies that attempt to analyze extrinsic motivation toward study in general, although we highlight the existence of some studies that have analyzed the relationship between intrinsic motivation and academic performance [21,22].

In line with these ideas, there are few, or practically nonexistent, studies that analyze this relationship with secondary school students. Some studies on future expectations in this population focused mainly on the relationship between future expectations and life satisfaction, self-esteem, and self-efficacy [50], with a significant and positive relationship between these variables. On the other hand, lower expectations were found in less socially adapted adolescents [51]. Data also showed that the perception of lack of family support was also associated with lower future expectations [52]. Regarding sex, some studies have found that boys showed lower future expectations than girls [50,52], and differences in future expectations were also found according to nationality and age [52], with older adolescents showing lower academic and economic labor expectations than younger adolescents. 

In our study, participants indicated the main sources of motivation to be having a test coming up, having an assigned homework, the fact that parents think studying is important, having the habit of always studying, being interested in the contents, or that classmates also study. Following the contributions of the self-determination theory (SDT) [53], we could identify that the main source of motivation of the participants was identified extrinsic motivation, where the motivation comes from wanting to get a good result rather than from the learning process itself (having a test in the next few days or having an assigned task are examples of this). We also identified controlled extrinsic motivation which concerns seeking the approval of others (studying because parents think it is important or that peers also study are examples of this). In short, motivation responded to some kind of imposition, either from the surrounding environment or internally. On the other hand, having the habit of always studying or being interested in the content could be considered sources of intrinsic motivation. Additionally, in this case, the participants manifested a motivation consistent with their behavior and thoughts [54]. 

As for future expectations, they expressed their intention to continue studying because they do not yet know what they want to do, to work because the position they want to perform does not require university studies, to work because they intend to be economically independent, or doing something else. Therefore, we found results that indicated a greater intention to continue their studies at higher levels. These results are similar to those found in other studies [44]. In addition, the future expectations of the students were influenced to a great extent by their own training or education options, being good in some specialty, talent, economic support available to them, academic performance, social status, leisure, the opinion of their parents, and the plans of their friends. In this sense, the results indicated a greater influence of personal issues (such as one’s own training or talent) above other issues of an economic or social nature or the opinions of people close to them. However, these results are not consistent with [55] in which parental opinion in decision making regarding the academic future of their children was most important. In the same direction, the results of [48] stands out.

Another objectives of this study was to develop a predictive model for future expectations as a function of variables such as sex, grade repetition, the reasons that encourage students to study or not to study (not being interested in the content and never studying), and the influences on expectations (school grades and subject mastery). The model explained 20.9% of the variability of future expectations of high school students based on variables such as grade repetition, not being interested in the content, and never studying, as well as school grades and being good at a specialty. However, in our study, sex did not predict future expectations, although some significant differences due to sex were found, similar to other studies with adolescent samples where boys had lower future expectations than girls [13]. With regard to the relationship between attitudes toward studying and future expectations, we have not found any studies that analyze this relationship, so data from the present study are pioneering in this regard. On the other hand, previous studies highlighted that family accompaniment in schoolwork could positively contribute to increasing future expectations in adolescence [51,52,53,54,55]. However, our study contemplated a greater number of variables in studying the influence of situations or people in shaping the future expectations of adolescents (school grades, subject mastery, and being talented).

The last major objective of this study was to confirm a hypothetical model of future expectations based on variables such as grade repetition, the reasons that encourage not studying, and the influences on future expectations. The results of this structural equation model showed that grade repetition had a negative influence on the reasons for not studying (not showing interest in the contents, nobody telling them to study, classmates not studying, and never studying) and these in turn positively influenced future expectations, which were also positively influenced by school grades, being good at a specialty, and having talent. Previous studies found relationships between the motivation to study and academic performance in high school students [55,56,57,58], supporting the results obtained in this study. In this study, grade repetition predicted the reasons that do not encourage studying, the results being consistent with those provided in other studies where repetition contributed to school dropout and had negative repercussions on an emotional level [31,32,34].

It is undeniable the great weight that motivation has in the psychological and social adjustment of adolescents, so different types of public policies to improve both intrinsic and extrinsic motivation of students in compulsory secondary education are essential, and currently we found more problems of demotivation than learning [21]. Therefore, attention to the diversity of high school students should be improved through creative methodologies that stimulate different intelligences in an attempt to increase intrinsic motivation that can significantly change habits and behaviors; on the other hand, it is essential to contribute to the promotion of students by creating and designing different educational itineraries that enhance their abilities and talents which necessitates an adequate academic and professional orientation starting from the first cycle of secondary education and even in earlier school stages.

Therefore, student motivation should become a primary objective of educational systems. One of the main protagonists is the teachers themselves who are responsible for ensuring that students have a good predisposition towards education and a positive attitude [45,50,59,60]. In short, the students’ achievements, results, attitudes, and the overcoming of their difficulties are highlighted in an educational environment where acceptance and a feeling of belonging reign, affecting students’ motivations [19]. Consequently, it is necessary to facilitate and contribute to the continuous training of teachers, mainly in the management and application of innovative methodologies, to increase the motivation of students as well as to reinforce and strengthen academic guidance from the daily scenario of educational institutions. Finally, this study is not without limitations. Among others, its cross-sectional nature makes it advisable for the data to be replicated by longitudinal studies, which would allow the results to be followed up over time.

## Figures and Tables

**Figure 1 ijerph-19-03864-f001:**
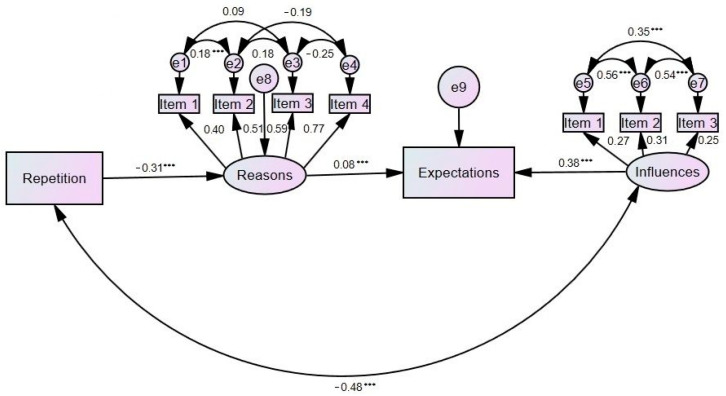
Results of the prediction of future expectations based on grade repetition, the reasons that do not encourage studying, and the influences on future expectations. Note: *** *p* < 0.001.

**Table 1 ijerph-19-03864-t001:** Contingency tables and chi-square test. Reasons that do not encourage studying and grade repetition.

		Repeated a Grade		Did Not Repeat a Grade		χ^2^	df	*p*	V
Motives that Do Not Encourage Study		Frequency	Corrected Residuals	Frequency	Corrected Residuals				
Not being interested in the content	Yes	527	8.0	1481	−8.0	64.00	1	0.000	0.110
	No	559	−8.0	2714	8.0				
Classmates do not study	Yes	357	12.2	710	−12.2	147.72	1	0.000	0.169
	No	680	−12.2	3395	12.2				
Nobody tells them that they have to study	Yes	376	10.2	846	−10.2	103.98	1	0.000	0.141
	No	694	−10.2	3312	10.2				
Never studying	Yes	411	15.8	648	−15.8	250.51	1	0.000	0.220
	No	644	−15.8	3428	15.8				

**Table 2 ijerph-19-03864-t002:** Contingency table and chi-square test. Reasons that encourage study and grade repetition.

		Repeated a Grade		Did Not Repeat a Grade		χ^2^	df	*p*	V
Reasons for Study		Frequency	Corrected Residuals	Frequency	Corrected Residuals				
Being interested in the content	Yes	30,058	23.1	6922	−23.1	535.50	1	0.000	0.142
	No	3018	−23.1	13,474	23.1				
Having an exam nearby	Yes	4965	−17.1	18,570	17.1	290.84	1	0.000	0.104
	No	1191	17.1	2306	−17.1				
Classmates do study	Yes	2195	21.5	4643	−21.5	463.73	1	0.000	0.133
	No	3769	−21.5	15,647	21.5				

**Table 3 ijerph-19-03864-t003:** Contingency table. Future expectations and grade repetition.

	Repeated a Grade		Did Not Repeat a Grade	
Future Expectations	Frequency	Corrected Residuals	Frequency	Corrected Residuals
Working because the occupation I want does not require a study degree	1606	32.9	1929	−32.9
Working because I need to be financially independent	1606	33.1	1678	−33.1
Studying because I do not know what I would like to do yet				
Studying because the occupation I want requires a study degree	1640	−53.8	13,764	53,8
Studying or working for other reasons	991	−14.2	1923	−14.2
Doing something else	349	−12.8	482	−12.8

**Table 4 ijerph-19-03864-t004:** Contingency table. Future expectations and sex.

	Girls		Boys	
Future Expectations	Frequency	Corrected Residuals	Frequency	Corrected Residuals
Working because the occupation I want does not require a study degree	1284	−17.9	2254	17.9
Working because I need to be financially independent	1285	−11.9	1889	11.9
Studying because I do not know what I would like to do yet	1976	−2.9	2085	2.9
Studying because the occupation I want requires a study degree	8808	24.0	6600	−24.0
Studying or working for other reasons	1385	−3.4	1532	3.4
Doing something else	361	−4.2	474	4.2

**Table 5 ijerph-19-03864-t005:** Contingency table. Influences on future expectations and grade repetition.

		Not Important		Somewhat Important		Important		Very Important		χ^2^	df	*p*	V
Influences on Future Expectations		Frequency	Residuals	Frequency	Residuals	Frequency	Residuals	Frequency	Residuals				
Grades	Yes	378	13.7	1045	13.7	2884	2.1	2224	−15.3	468.62	3	0.000	0.128
	No	536	−13.7	2255	−13.7	9487	−2.1	9927	15.3				
Subject mastery	Yes	359	14.4	1052	16.8	3166	0.4	1919	−16.0	613.474	3	0.000	0.146
	No	469	−14.4	1976	−16.8	10,735	−0.4	8981	16.0				
Friends	Yes	1351	−20.2	2441	−0.5	2164	16.0	560	11.1	586.93	3	0.000	0.143
	No	7514	20.2	8383	0.5	5191	−16.0	1097	−11.1				
Having a talent	Yes	331	11.6	1052	13.6	3124	1.7	1968	−14.7	429.687	3	0.000	0.123
	No	516	−11.6	2239	−13.6	10,419	−1.7	8962	14.7				

**Table 6 ijerph-19-03864-t006:** Contingency table and chi-square test. Attitude towards the study and expectations of future.

		Working Occupation Not Require a Study Degree		Working Need to Be Financially Independent		Studying Do Not Know What I Would Like to Do yet		Studying Occupation Requires a Study Degree		Studying or Working		Doing Something Else		χ^2^	*p*	V
Attitude before the Study		Frequency	Corrected Residuals	Frequency	Corrected Residuals	Frequency	Corrected Residuals	Frequency	Corrected Residuals	Frequency	Corrected Residuals	Frequency	Corrected Residuals			
Not being interested in the content	Yes	246	5.2	215	4.3	313	2.8	903	−10.6	205	4.0	69	2.4	117.14	0.000	0.151
	No	262	−5.2	243	−4.3	427	−2.8	1974	10.6	236	−4.0	77	−2.4			
Classmates do not study	Yes	153	5.9	119	3.6	156	0.9	447	−9.3	116	3.7	44	3.0	103.72	0.000	0.143
	No	346	−5.9	320	−3.6	559	−0.9	2381	9.3	307	−3.7	100	−3.0			
Nobody tells them that they have to study	Yes	172	6.1	142	4.4	195	2.5	502	−10.5	125	2.9	47	2.6	119.36	0.000	0.153
	No	336	−6.1	309	−4.4	536	−2.5	2353	10.5	311	−2.9	1183	−2.6			
Never studying	Yes	168	7.3	157	7.8	155	0.6	412	−12.5	124	305	42	101	195.97	0.000	0.197
	No	334	−7.3	289	−7.8	560	−0.6	2423	12.5	4.3	−4.3	2.5	−2.5			
Classmates do study	Yes	1047	12.3	900	9.4	951	2.0	2878	−18.1	676	2.3	235	5.5	407.68	0.000	0.126
	No	1926	−12.3	1791	−9.4	2536	−2.0	10,692	18.1	1754	2.3	434	−5.5			

**Table 7 ijerph-19-03864-t007:** Multinomial regression analysis for future expectations based on different predictor variables.

Future Expectations		B(DE)	Odds Ratio	95% CI for OR	
				Inferior	Superior
Working because the occupation I want does not require a study degree	Intercept	1.20 (0.28) ***			
	Sex (Female)	−0.12 (0.20)	0.88	0.58	1.33
	Grade Repetition (Did not repeat a grade)	0.07 (0.21)	1.08	0.71	1.63
	Not being interested in the content (Yes)	0.07 (0.21)	1.07	0.70	1.64
	Never studying (Yes)	0.32 (0.24)	1.38	0.86	2.23
	Grades (Not important)	−0.62 (0.50)	0.53	0.19	1.44
	Grades (Somewhat important)	−0.49 (0.33)	0.61	0.31	1.18
	Grades (Important)	−0.11 (0.27)	0.89	0.52	1.53
	Subject mastery (Not important)	0.65 (0.53)	1.91	0.67	5.46
	Subject mastery (Somewhat important)	0.77 (0.39) *	2.17	1.00	4.68
	Subject mastery (Important)	−0.05 (0.27)	0.94	0.55	1.60
Working because I need to be financially independent.	Intercept	1.30 (0.28) ***			
	Sex (Female)	−0.02 (0.21)	0.97	0.64	1.47
	Grade Repetition (Did not repeat a grade)	−0.16 (0.21)	0.84	0.55	1.28
	Not being interested in the content (Yes)	0.03 (0.21)	1.03	0.67	1.58
	Never studying (Yes)	0.41 0(.24)	1.51	0.93	2.45
	Grades (Not important)	−1.16 (0.54) *	0.31	0.10	0.91
	Grades (Somewhat important)	−0.30 (0.34)	0.74	0.37	1.44
	Grades (Important)	0.12 (0.27)	1.13	0.65	1.95
	Subject mastery (Not important)	0.22 (0.55)	1.25	0.41	3.74
	Subject mastery (Somewhat important)	0.34 (0.39)	1.41	0.65	3..08
	Subject mastery (Important)	−0.28 (0.27)	0.74	0.44	1.27
Studying because I do not know what I would like to do yet	Intercept	1.09 (0.28) ***			
	Sex (Female)	0.28 (0.20)	1.33	0.90	1.97
	Grade Repetition (Did not repeat a grade)	1.10 (0.21) ***	3.01	1.98	4.57
	Not being interested in the content (Yes)	0.05 (0.20)	1.05	0.70	1.58
	Never studying (Yes)	0.02 (0.24)	1.02	0.63	1.63
	Grades (Not important)	−1.19 (0.53) *	0.30	0.10	.86
	Grades (Somewhat important)	−0.62 (0.32) *	0.53	0.28	1.01
	Grades (Important)	−0.02 (0.26)	0.98	0.58	1.64
	Subject mastery (Not important)	−0.59 (0.57)	0.55	0.17	1.70
	Subject mastery (Somewhat important)	0.12 (0.38)	1.13	0.53	2.42
	Subject mastery (Important)	−0.32 (0.25)	0.72	0.44	1.19
Studying because the occupation I want requires a study degree	Intercept	2.15 (0.26) ***			
	Sex (Female)	0.34 (0.19)	1.41	0.97	2.06
	Grade Repetition (Did not repeat a grade)	1.97 (0.20) ***	7.18	4.84	10.66
	Not being interested in the content (Yes)	−0.26 (0.19)	0.76	0.51	1.12
	Never studying (Yes)	−0.16 (0.23)	0.84	0.53	1.32
	Grades (Not important)	−1,71 (0.49) ***	0.17	0.06	0.46
	Grades (Somewhat important)	−1,17 (0.31) ***	0.31	0.16	0.57
	Grades (Important)	−0.34 (0.25)	0.70	0.43	1.15
	Subject mastery (Not important)	−0.81 (0.52)	0.44	0.15	1.24
	Subject mastery (Somewhat important)	−0.26	0.76	0.37	1.58
	Subject mastery (Important)	−0.49 (0.24) *	0.61	0.38	0.98
Studying or working for other reasons.	Intercept	0.81 (0.29) **			
	Sex (Female)	0.20 (0.21)	1.23	0.81	1.85
	Grade Repetition (Did not repeat a grade)	0.48 (0.21) *	1.61	1.05	2.47
	Not being interested in the content (Yes)	0.08 (0.21)	1.08	0.70	1.66
	Never studying (Yes)	0.31 (0.24)	1.37	0.84	2.23
	Grades (Not important)	−0.71 (0.54)	0.48	0.16	1.41
	Grades (Somewhat important)	−0.42 (0.34)	0.65	0.33	1.28
	Grades (Important)	0.11 (0.27)	1.12	0.65	1.93
	Subject mastery (Not important)	−0.25 (0.58)	0.77	0.24	2.41
	Subject mastery (Somewhat important)	0.24 (0.40)	1.27	0.57	2.79
	Subject mastery (Important)	−0.21 (0.26)	0.80	0.47	1.37

Note: R^2^ = 0.194 (Cox y Snell), 0.209 (Nagelkerke). Model *χ^2^*_(50)_ = 1042.79, * *p* < 0.05, ** *p* < 0.01 *** *p <* 0.001.

## Data Availability

Not applicable.

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
