# Peer review of "Motivational Profile, Future Expectations, and Attitudes toward Study of Secondary School Students in Spain: Results of the PISA Report 2018"

_ijerph, 2022, doi:10.3390/ijerph19073864_

Round 1

Reviewer 1 Report

This is an interesting study with important results. However, this reviwer believes that the statistical approach is soft and the results are very hard to read. The manuscript requires improvement. 

The introduction is well written. However it mainly focus intrinsic motivation. It would be better to explain how and what type of extrinsic motivation is possible to describe in students. It is possible that literature reports mainly intrinsic motivation. I recommend to better explore the extrensic cases (one paragraph).

Methods

Provide ethical aprovement information.

Instrument: why only some items? The authors may explain and justify based on previous studies with similar procedures.

Statistical analysis: was not possible to run confirmatory models? Structural equation modeling?

Results:

Too many text is hard to follow. Please provide figures, graphs or tables.

Discussion: hard to follow because the results are no easy to read.

It would be better to explain how these results may help to define public policies and what type of to increase school motivation. 

Reviewer 2 Report

Dear Authors, thank you very much for your efforts and research on this topic. I am glad to read and review your paper.

Introduction: Lines 36-37 "the search for employment, abandoning academic training" is this applicable for all participants or only for those who might have more finantial difficulties?

Lines 65-67 :"However, with respect to the type of motivation that predominates in second ary school students, several investigations, with Spanish samples, show that secondary school students are [18, 19, 20]." are what? The sentence needs an ending.

Materials and Methods: maybe the full description of language spoken (at home, parents, friedns and so on) can be summairzed to some key aspects.

In data analysis: the level of significance considered as cut-off for the analysis is missing, as well the cut-off for correlation analysis.

Results: lines 172-176 authors only reported that a statistical significance for the Chi-square tests were observed. It would be very informative to know how they differ, for example are boys less motivated than girls? Maybe a table with means and the results from statistical tests would be helpful to visualize the differences.

A table for the model coefficients (mutinomial regression model) would be also interesting to analyze.

Author Response

Thank you very much for your positive feedback and for your suggestions to improve this work. Below, we detail the changes made following your remarks:

Abstract

- Information on the new data from the regression model and on the results of the structural equation model is included.

Introduction

- In lines 36-37 the information is expanded and clarified, as suggested.

- The wording in lines 65-67 is also completed, following the recommendations.

- A paragraph is included in the introduction to refer the relationship between motivation and academic performance and on previous studies referring to future expectations to discuss the results.

- The country of birth and languages spoken variables are also eliminated from the study objectives and the work hypotheses are formulated.

Method

In the data analysis, the statistical analyzes are redrafted, based on the new regression analysis and the structural equation model.

Resultados

Results

- All information referring to variables such as country of birth and languages spoken is eliminated because they are not taken into account in the regression models and structural equations.

- All the tables on the descriptive and relational analysis (corrected residuals, Chi-square value and Cramer's V value) are included, following the correct contributions, and the previous wording of results is replaced by new approaches in the statistical analysis of data.

- A new multinomial regression model is included to explain future expectations and its corresponding table, as suggested.

- Tables are used to summarize the results, as suggested.

Discussion

- Added several paragraphs to discuss the new regression model and SEM results, and removed the old regression model results.

- The discussion is adapted to the objectives and hypotheses of the study to follow a common thread.

Reviewer 3 Report

 Motivational Profile, Future Expectations and Attitude Towards Study of Secondary School Students in Spain: Results of the PISA Report 2018

 The aim of the present work is to study the sources of motivation to study and the future expectations of secondary school students, and to develop a predictive model of their future expectations based on a number of variables studied. 13 The sample consisted of a total of 35,943 students from different Spanish high schools, with age of 15.83 (SD=.28). The instrument used were the placement tests referring to the PISA 2018 Report. The results showed that the main source of motivation of secondary school students to study responds to some kind of imposition, either from the surrounding environment or internally, which appears represented by an identified or controlled extrinsic motivation. In terms of future expectations, their intention was to continue studying , before they influences by other aspects , such as economic support for it, and the opinion of parents and friends.  Based on these results, some advice and educational implications are discussed

 The paper analyses PISA 2018 Report data and this is the strong point, considering the huge sample and the measured variables.

 The data analysis was based on descriptive statistics and the Chi-Square test and a multinomial logistic regression analysis.

The results are not remarkable in the sense that are rather anticipated, however the value is on the specific PISA data set, which are worth analyzing and reporting

An issue here arises from the bivariate statistical test and the statistical significance in all cases, which might be artifact of the huge sample. Chi-Square test is not recommended here. Cramer’s V might be used or other statistics. Please discuss this issue.

I propose to present effect sizes.

Moreover, the results presented in a narrative way is rather hard to follow, I recommend to use Tables to summarize the statistics and the finding.

Author Response

Thank you very much for your positive feedback and for your suggestions to improve this work. Below, we detail the changes made following your remarks:

Abstract

- Information on the new data from the regression model and on the results of the structural equation model is included.

Introduction

- A paragraph is included in the introduction to refer the relationship between motivation and academic performance and on previous studies referring to future expectations to discuss the results.

- The country of birth and languages spoken variables are also eliminated from the study objectives and the work hypotheses are formulated.

Method

- In the participants section, the information referring to the languages spoken is eliminated.

- In the instruments section, all the items referring to the variables studied are included (attitude towards study, future expectations, influences on future expectations) and information referring to the variables country of birth and languages spoken is suppressed.

Results

- All information referring to variables such as country of birth and languages spoken is eliminated because they are not taken into account in the regression models and structural equations.

- All the tables on the descriptive and relational analysis (corrected residuals, Chi-square value and Cramer's V value) are included, following the correct contributions, and the previous wording of results is replaced by new approaches in the statistical analysis of data.

- Tables are used to summarize the results, as suggested.

Discussion

- Added several paragraphs to discuss the new regression model and SEM results, and removed the old regression model results.

Round 2

Reviewer 1 Report

The authors addressed my concerns and the manuscript is easier to read.